# Postnatal and Adult Neurogenesis in Mammals, Including Marsupials

**DOI:** 10.3390/cells11172735

**Published:** 2022-09-01

**Authors:** Katarzyna Bartkowska, Beata Tepper, Krzysztof Turlejski, Ruzanna Djavadian

**Affiliations:** 1Nencki Institute of Experimental Biology, Polish Academy of Sciences, 02-093 Warsaw, Poland; 2Faculty of Biology and Environmental Sciences, Cardinal Stefan Wyszynski University in Warsaw, 01-938 Warsaw, Poland

**Keywords:** marsupials, eutherians, developmental neurogenesis, adult neurogenesis, dentate gyrus, olfactory bulb, opossum, *Monodelphis domestica*

## Abstract

In mammals, neurogenesis occurs during both embryonic and postnatal development. In eutherians, most brain structures develop embryonically; conversely, in marsupials, a number of brain structures develop after birth. The exception is the generation of granule cells in the dentate gyrus, olfactory bulb, and cerebellum of eutherian species. The formation of these structures starts during embryogenesis and continues postnatally. In both eutherians and marsupials, neurogenesis continues in the subventricular zone of the lateral ventricle (SVZ) and the dentate gyrus of the hippocampal formation throughout life. The majority of proliferated cells from the SVZ migrate to the olfactory bulb, whereas, in the dentate gyrus, cells reside within this structure after division and differentiation into neurons. A key aim of this review is to evaluate advances in understanding developmental neurogenesis that occurs postnatally in both marsupials and eutherians, with a particular emphasis on the generation of granule cells during the formation of the olfactory bulb, dentate gyrus, and cerebellum. We debate the significance of immature neurons in the piriform cortex of young mammals. We also synthesize the knowledge of adult neurogenesis in the olfactory bulb and the dentate gyrus of marsupials by considering whether adult-born neurons are essential for the functioning of a given area.

## 1. Introduction

In eutherians, the generation of new neurons in the central nervous system (CNS) and the formation of almost all brain structures occur during embryonic development, known as developmental neurogenesis. It is now well established that new neurons are continuously produced in adult mammalian brains, and this process is known as adult neurogenesis. Unlike eutherians, marsupials are born immature, and most brain structures that arise from the forebrain are formed after birth, during the postnatal period, and this is also classified as developmental neurogenesis (for review, see [1]). In eutherians, the generation of granule cells in the olfactory bulb (OB) and dentate gyrus (DG) occurs postnatally, although the formation of these structures starts during embryogenesis [2,3]. Similar developmental events are observed in the cerebellum, where neurogenesis begins to take place during embryonic development and persists after birth, for a few weeks or months, depending on the brain size and the stage of cerebellar development [4,5].

In the 1960s, Altman and colleagues reported that new neurons are born in the DG (mostly known as adult hippocampal neurogenesis) and the subventricular zone of the lateral ventricle (SVZ) [6,7]. These findings were contrary to Ramón y Cajal, who suggested that, in mammals, new neurons could not be added to the CNS after birth [8]. New approaches, including using 5-bromo-2-deoxyuridine (BrdU) as a synthetic analog of thymidine, and molecular markers for specific cell types, allow studying this problem again. It has been established that new neurons generated in the DG migrate a short distance from the subgranular layer to the granular layer of the DG and remain within this structure [9]. However, the majority of proliferated cells in the SVZ migrate long distances and reside in the OB, which is the final destination for new neurons [10]. Active adult neurogenesis in the SVZ and the DG lasts through life but declines with age [11,12]. The existence of adult neurogenesis in humans and its functions are currently being discussed [13]. The controversy is also related to whether adult neurogenesis occurs in brain structures such as the piriform cortex, cerebral cortex, hypothalamus, striatum, amygdala, and others [14,15]. Recently, in one eutherian species of the order Lagomorpha, the New Zealand white rabbit, newborn neurons were detected in the adult cerebellum [16]. The question, therefore, arises whether the cerebellum can be considered another neurogenic brain structure.

Here, we will highlight the postnatal development of certain brain structures in mammals, including the postnatal generation of granule neurons, specifically focusing on the similarities and differences in developmental neurogenesis between marsupials and eutherians. We will also synthesize the knowledge about adult neurogenesis in marsupials and eutherians, considering the structures for which data on adult-born neurons in marsupials are available.

## 2. Postnatal Developmental Neurogenesis in Mammals

The mammalian subclass Theria contains two infraclasses: the Metatheria, or marsupials, and the Eutheria, or placental mammals. The split of the two infraclasses, Marsupialia and Eutheria, is postulated to have occurred at least 160 million years ago. Notably, the discovery of a new fossil record related to a new eutherian species, *Juramaia sinensis*, allowed re-setting the time for the diversification of marsupials and eutherians [17]. Living marsupials are divided into two main groups, American marsupials (Ameridelphia) and Australian marsupials (Australidelphia), and approximately 70% of marsupial species are found in Australia.

One of the main differences between marsupials and eutherians is in the reproduction process. The gestation period of all marsupials is short, ranging from 12 days to 5 weeks [18]. Marsupials give birth to immature progeny, and after birth, newborn marsupials attach to the mother’s nipples for a long time [19]. Some marsupial species, mostly representatives of the family Didelphidae, are pouchless, and newborns remain attached to the mother’s nipples on the lower abdomen (Figure 1), while, in others, this area is covered with a skin fold called a pouch.

Marsupials are born small, hairless, and immature, and their brain is in a very early stage of development. The development of the hindbrain is accelerated to match the accelerated development of the structures of the mouth, face, and front limbs, used by newborn opossums to travel and attach to the mother’s nipples and suck milk [20]. In other brain structures, including the midbrain and the forebrain, intense neurogenesis occurs after birth [1,4,21,22,23,24]. In general, marsupials are born at a developmental stage similar to the mid-embryonic stage of eutherians; however, different degrees of morphological development are observed among marsupials. For example, newborn possums (*Trichosurus vulpecula*) have external ears and eye primordia, whereas the heads of newborn dasyurids are composed of the nose and mouth [20]. As there are some differences in development between marsupial species, we will provide data on the brain development of the gray short-tailed opossum *Monodelphis domestica* (Ameridelphia) and a representative of Australian marsupials, the tammar wallaby (*Macropus eugenii),* recently called *Notamacropus eugenii*. Data on the brain development of marsupials are scarce, and these species have been chosen because the development of some brain structures has been best studied in both of these marsupials. We compared the selected brain structures development of opossums to that of mice (*Mus musculus*) or rats (*Rattus norvegicus*) as representatives of eutherians, which are laboratory animals with smooth brains.

Apart from this, the gray short-tailed opossum is a small animal (60–150 g) with a smooth brain, while the tammar wallaby is a medium-sized animal (4–10 kg) with a relatively convoluted brain surface. The full characteristics of these marsupials are the following.

The opossum, *Monodelphis domestica,* is a pouchless marsupial. The name “domestica” comes from the fact that opossums often inhabit human dwellings. They are omnivores, feeding mainly on small vertebrates, invertebrates, and fruits. After 15 days of gestation, newborn opossums weighing only 100–120 mg are born [25]. They are at a developmental stage comparable to the rat embryonic (E) stage at day 12 or a 6-week embryo in humans [26,27]. During the 3 weeks after birth, almost all brain structures of the opossum are formed with different types of neurons, except granule cells of the olfactory bulb and the cerebellum [28,29]. The 2-month-old opossum already has a developed brain, and this is the age of weaning. At the age of approximately 5 months, opossums are sexually mature, and they live up to 3 years. Females are smaller (weighing 60–90 g) than males (90–150 g). They are solitary animals; therefore, in laboratory colonies, each adult opossum is kept individually in a separate cage.

The tammar wallaby is a nocturnal, herbivorous species, and its diet consists mainly of grasses and shrubs. They live in groups and have seasonal breeding. The tammar wallaby is born after a short gestation period (27 days of pregnancy) with an underdeveloped brain, whose developmental stage is equivalent to that of a newborn *Monodelphis domestica* opossum [30]. The body weight of a newborn tammar wallaby is approximately 400 mg. After birth, the newborn wallaby climbs into the pouch, attaches to the mother’s teat, and remains there before reaching maturity [30]. The tammar wallaby grows and develops in the pouch over the next 8–9 months, which is also associated with an extended time of brain development. Females and males sexually mature at the age of about 9 months and 2 years, respectively. The average body weight of adult tammar wallabies is 4–10 kg, and they live 11–14 years.

The mouse gestation period is 20 days, and the average litter size is 8, with weaning at 21–26 days [31]. At birth, newborn mice body mass (1.25 g) is 10-fold greater than that of opossums (*Monodelphis domestica*), while adult mice (25–35 g) weigh 3.5–4 times less than adult opossums.

In both marsupial species, the opossum and the tammar wallaby, the hypothalamus develops postnatally, while in the mouse, it develops embryonically [32,33,34,35]. In the wallaby, injections of ^3^Hthymidine revealed that neurons started to generate at P12 to P25 [34]. In the opossum, the development of the hypothalamic suprachiasmatic nuclei has been studied using the ^3^Hthymidine autoradiography method [35]. A rostrocaudal gradient of neurogenesis was determined in the developing hypothalamus. Newborn neurons were found in the suprachiasmatic nuclei at P3 opossums, and their numbers gradually decreased until P7 in the anterior region of the hypothalamus, while neurogenesis was completed in the caudal hypothalamic region at P10. ^3^Hthymidine injections to mice at different days of pregnancy showed that heavily labeled cells in the hypothalamic nuclei were observed at embryonic days 10 to 16 [32]. These data indicate that in mice, the formation of the hypothalamic nuclei takes place over 6 days.

The neocortex is the main brain structure of the mammalian cerebral cortex and varies extremely in size and shape across species. Nonetheless, in all mammals, the neocortex consists of six layers; therefore, it is also called the six-layered cortex. The cells of the cerebral cortex are generated by neural progenitors located in the ventricular zone (VZ) of the brain [36,37]. The newly generated neurons migrate from the VZ/SVZ and create the six-layered mammalian cortex. Projection neurons of the deep layers (VI/V) are generated first, send their axons to the subcortical structures (spinal cord, pons, thalamus), and form connections there, while neurons of the upper layers (III/II) make connections between different cortical areas in the same or contralateral hemispheres [38]. In marsupials, the major commissure connecting the neocortex of the opposite hemispheres is the anterior commissure. In contrast, eutherians evolved a new commissure called the corpus callosum [38].

The neocortex of the opossum is less fissured than that of the tammar wallaby. However, in both the newborn opossum and wallaby, the neocortex has similar organization; it is extremely immature, containing two layers, the primordial plexiform and ventricular layer, which are also seen in the embryonic neocortex of eutherians [26,30,39]. Three days after birth, the cortical plate begins to appear in the telencephalon of the opossum brain, while this process is protracted in the wallaby. In this species, the cortical plate starts to differentiate at P10 and P15, and during this period, the SVZ increases, and the subplate layer appears. During the following 50 days, between P20 and P70, a 6-layered cortex is created, following the pattern observed in eutherians. Time-course analysis using molecular markers for specific layers of the neocortex in the gray short-tailed opossum showed that neurons are generated between P1 and P16 [40]. However, data on BrdU labeling of cortical cells in P17 opossum brains demonstrated a rostroventral to caudodorsal gradient in the neocortex, and new neurons were still being produced in the upper layers of the caudal neocortical region [41,42]. To determine the organization of mice neocortex, ^3^H thymidine injections have been performed on pregnant mice on different days of gestation [43]. The results of these experiments showed that the production of neurons forming the neocortical layers started at E12.5 and continued up to E18.5. Overall, in mice, the whole cortical plate, including layers VI-II, is formed within 6 days [44], while in opossums, all the cortical layers develop over 15 days.

Two brain structures, the OB and DG, are unique. The formation of both brain structures in eutherians starts during the embryonic period, and most of the granule cells develop postnatally, although their functions are different. Granule cells in the DG are glutaminergic, while granule cells of the OB are GABAergic. Their similarity is only in size; that is, they are small neurons. Comparative studies have demonstrated that the DG organization is similar within different mammalian species [45,46]. The DG is a trilaminar brain structure; the soma of granule neurons is located in the middle granule layer, and their dendrites form the outer molecular layer. The third layer is a hilus (also called a polymorphic layer) that contains a number of cell types, including mossy cells [47]. Altman and Das [6] were the first to report that granule cells in the DG of the rat are generated postnatally with a peak at P15. In the same year, a paper by Angevine [48] demonstrated the origin of the granule cells in the DG of another rodent species, showing that in mice, the generation of these cells starts at E10 and continues until P20. The most recent studies on neurogenesis in the developing mouse brain have reported that neurogenesis in the DG begins at E12.5 and decreases at P14 [49]. There are no data on neurogenesis in the developing DG of the opossum and tammar wallaby. However, research carried out in another representative of marsupials, the quokka (*Setonix brachyurus*), has shown that a few cells labeled with ^3^Hthymidine were seen in the hilus of the DG at P5 [50]. A peak of these labeled cells was observed at P40, whereupon their number gradually decreased. At P85, the oldest age of quokka examined in this paper, cell generation was still going on. The quokka wallaby weighs between 2.5 and 5 kg, which is half the size of the tammar wallaby, and has a gestation period of 28 days, which is very similar to that of the tammar wallaby [50].

A comparative analysis of the DG development in eutherians reveals that this process is prolonged compared to other brain areas. For example, the development of the DG and the neocortex in the mouse starts at the same developmental period (around E12) [44]. However, neurogenesis in the neocortex ceases earlier than the generation of neurons in the DG. Later, after having completed the neocortex development, the peak of neurogenesis occurs in the DG [6,48]. We suggest that this protracted development of the DG in eutherians can be related to brain structural connectivity. The main input to the hippocampus comes from the neocortex. Therefore, the neocortex develops first, which then provides projections to the DG.

In marsupials, the DG and the neocortex begin to develop simultaneously and continue to proceed over time until their development is complete [22,50]. This indicates that both brain structures follow a similar developmental time course. Since this observation is based only on data from one paper, further work is needed on various marsupial species to understand the significance of different developmental time windows for specific brain areas.

The olfactory system contains olfactory receptor neurons, whose axons send information about various odors to the OB. In mice, there are over 1400 different types of olfactory receptor neurons, each expressing only 1 type of olfactory receptor, and all receptor neurons of the same type send axons to a specific glomerulus of the OB (for review, see [51]). Dendrites of mitral and tufted cells form synapses with projecting axons of the olfactory receptor cells. Glomeruli are composed of neuropil, which is surrounded by juxtaglomerular neurons. The soma of mitral cells creates the next layer, called the mitral cell layer. Below this layer, the mitral and tufted cells, axons, and dendrites of granule cells form the internal plexiform layer. The deepest granule cell layer consists of granule neurons [52]. In the mice OB, projecting neurons, both mitral and tufted cells, are generated during embryonal neurogenesis from E11 to E18. Some interneurons and periglomerular cells are proliferated before birth, while most granule cells and periglomerular cells are largely generated during early postnatal life, over 20 days [53]. Although mitral and tufted cells are generated during embryonal development, their maturation occurs postnatally; therefore, the OB layered organization becomes apparent after birth in mice [53].

The cellular organization of the OB in the developing opossum and the tammar wallaby has been studied using histological techniques [28,54]. The neuroepithelium of the OB in the tammar wallaby is increased in thickness at P5, but the first postmitotic neurons appear in the ventromedial region of the OB only by P12. Glomeruli organization is determined by P25, and better lamination is seen at P37. Accordingly, the OB is highly organized in the opossum. It has been demonstrated that this brain region of the newborn opossum contains mainly neuroepithelium and a few neurons, and neurogenesis continues till P30 [28].

In marsupials, the sequence of developmental events and the general pattern of brain development are very similar to those of eutherians. However, there are also notable differences in the developmental time course between them. In marsupials, most brain structures develop postnatally and have a protracted timing of development, while the same brain structures of eutherians develop embryonically in a short developmental time window. For comparison, neocortical neurogenesis in the mouse occurs within 5 days (E12.5 to E17.5), while the development of neocortical layers in the opossum occurs mainly postnatally and lasts 15 days (P3 to P18) [41,42,44]. There are, however, some exceptions to this. In particular, the hypothalamus develops postnatally in the opossum and prenatally in the mouse, but with a similar developmental time window (about 6 days) [32,35]. We suggest that a similar developmental timing of the hypothalamus in marsupials and eutherians is linked with its function. The hypothalamus is responsible for essential functions, such as regulation of thirst or sleep; therefore, it develops in a short developmental time window, even in marsupials. In turn, the neocortex is associated with higher-level processes; as the overall development of the opossum is slow, the neocortex development is also protracted.

### Postnatal Neurogenesis in the Cerebellum

In eutherian mammals, the cerebellum is a unique brain structure where the generation of new neurons starts in the early embryonic stage and continues postnatally. The cerebellum is a structure that consists of two main parts, the cerebellar deep nuclei (DCN) and the cerebellar cortex. The DCN are output structures that receive input from the neurons of the cerebellar cortex. In turn, the cerebellar cortex is made up of different types of cells that are arranged in a very organized manner, forming a 3-layered structure of the cerebellar cortex, first described by Cajal [55]. The following layers of the cerebellar cortex are distinguished: the outer molecular layer, the Purkinje cell layer, and the granular layer. The largest neurons of the brain, which are Purkinje cells, are found in the cerebellum. The most numerous neurons, which are cerebellar granule cells, are also found in the cerebellum. Apart from these two types of neurons, other cell types (basket, stellate, Golgi, and unipolar brush cells) are also present in the cerebellum.

Although mammalian species differ in the size of the cerebellum, they all share similar cellular and layered organization [56]. Eutherian species that are born with closed eyes have an immature cerebellum, whereas species born with open eyes have a relatively mature cerebellum [57]. All mammalian species first develop the DCN and Purkinje cells, followed by the proliferation and development of granule cells. In eutherians, this second phase of new neuron generation occurs after birth, during postnatal development, and this process refers to developmental neurogenesis, during which progenitor cells give rise to numerous cerebellar granule cells. Two distinct areas, the ventricular zone of the fourth ventricle and its rhombic lip, are known as neurogenic zones, from which neurons are generated and form the cerebellum [58,59]. Proliferated cells migrate from the rhombic lip and locate in the external germinal layer of the cerebellum, creating the second germinal zone of the cerebellum [60]. Almost all neurons in the granular layer are produced from progenitor cells that proliferate in the external germinal zone [61,62]. In rats, the generation of granule cells starts after birth and lasts until P21 [63], while in humans, the first granule cells proliferate during the 28th gestational week. The generation of granule cells is completed by the 11th postnatal month, with a peak within the 28th and 34th gestational weeks [64]. However, during the first 5 postnatal months, the rate of neurogenesis in the EGL occurs at relatively higher levels. By the 5th postnatal month, about 30% of EGL cells are still proliferative. The number of newly proliferated cells gradually declines in the following months, and eventually, this second germinal layer disappears in humans by the 11th postnatal month [64].

We also illustrate the development of the cerebellum in the rabbit (lagomorph), which is the first and only species so far investigated that shows adult cerebral neurogenesis. In the lagomorphs, the development of cerebellar granule cells is more protracted compared to that of rodents. The time-course of the cerebellum’s development has been examined in one of the species of the order Lagomorpha, the New Zealand white rabbit [16]. In the cerebellum of this species, newly generated neurons were observed in the cerebellum of 3-year-old rabbits [16]. At this age, rabbits are sexually mature and considered adults.

The newborn opossum is characterized by an extremely undeveloped cerebellum. All types of neurons in the cerebellum are generated postnatally [29]. Purkinje cells originate from progenitors residing in the VZ within postnatal days 1–5 (Figure 2A), while the proliferation of granule cells begins 3 weeks later, and intensive cell divisions are observed between P21 and P50 (Figure 2B,C), after which the rate of proliferation is ceased (Figure 2E–G).

Thus, a substantial number of granular layer neurons are generated after birth during postnatal development [29]. In opossums, the EGL disappears when animals are approximately 5 months old

Taken together, the considered studies indicate that the development of the cerebellum is conserved in both marsupials and eutherians. It is a highly complex process with similar sequences in the generation of different cerebellar types of neurons; in particular, the DCN and Purkinje cells are generated first, followed by granule cells, to establish the layering of the cerebellum. In all investigated species of mammals, granule cells are produced after birth in a limited period of cerebellar growth. In this species, newly born neurons persist in the adult cerebellum of the rabbit for unknown reasons.

## 3. Adult Neurogenesis in Mammals

A number of brain areas are considered to be proliferative zones in adults of mammalian species. Two areas, the SVZ and the dentate gyrus, are well-known neurogenic zones where the proliferation of cells occurs throughout life. According to available data on adult neurogenesis in mammals, new neurons are generated in the adult brain of opossums [65] and the fat-tailed dunnarts [66], rodents, including laboratory mice and rats [6,67,68,69], carnivores [70,71], sheep [72,73], shrews [74,75], giant otter shrews [76], tree shrews [77,78], the rock hyrax and sengi [79], hedgehogs and European moles [80], bats [81,82], and primates, including humans [83,84,85]. More and more evidence is currently accumulating about adult brain structures, such as the piriform cortex, cerebral cortex, corpus callosum, striatum, amygdala, and hypothalamus, where cells express different markers for cell proliferation (for review, see [16,86]). This indicates that cells can be added to these structures in adult brains. More data on adult neurogenesis in various mammalian species are presented in Table 1.

### 3.1. Adult Neurogenesis in the SVZ/OB

Progenitor cells leave the SVZ after division and migrate long distances. Numerous papers have reported that in various mammals, including marsupials, the majority of proliferated cells of the SVZ migrate through the rostral migratory stream (RMS) and mature in the OB. These data are presented in Table 1. The OB is a relay structure of the brain for the olfactory system. In the glomerulus of the OB, information is selectively modulated by GABAergic and dopaminergic interneurons forming the periglomerular inhibitory network. The second inhibitory network is formed by GABAergic granule cells of the OB. This two-layered lateral inhibition enhances odor discrimination to make information more specific, and then the axons of mitral cells send it to the olfactory cortex (for review, see [139]). In mice, progenitor cells of the SVZ undergo asymmetric divisions generating neuroblasts that migrate along the RMS to reach the OB [140]. Most of them differentiate into granule or periglomerular neurons using a neurotransmitter, either GABA (the majority) or dopamine [141].

A large number of studies have revealed that new neurons are critical for odor detection and discrimination and olfactory learning and memory [142,143,144]. Newly generated granule cells participate in odor–reward association [145]. The elimination of adult-born neurons or inhibition of the rate of neurogenesis in the OB affects odor detection and odor discrimination regardless of the methods used [146,147]. In addition, adult-born granule neurons are involved in the regulation of mitral cell activity; that is, adult-born young neurons incorporated into the neuronal circuit of the OB enhance odor-induced responses of mitral cells, improving their power to discriminate between odors [148]. This effect is diminished as adult-born neurons become mature.

However, our knowledge about the function of adult-born granule cells of the OB is far from complete. It has become obvious that there are two different populations of granule cells in the OB. One population of granule neurons of the OB is generated during the embryonic development of mice, while a second population of granule neurons, consisting of the majority, is generated in postnatal life [149]. Data related to the function of these neuron populations are controversial. On the one hand, Sakamoto et al. [149] have revealed that inhibition or activation of interneurons activity in the adult OB of genetically-manipulated mice does not affect behavioral tasks requiring olfactory perception. However, Takahashi et al. [144] have reported that granule cells derived from postnatal neurogenesis are required for odor detection. More recent research has demonstrated that preexisting neurons are involved in complex learned discrimination only, whereas adult-born neurons are engaged in both simple and complex learned discrimination [150].

In humans, the presence of newborn neurons in the SVZ/OB remains controversial. First, Eriksson et al. [85] described the progenitor cells in the SVZ adjacent to the caudate nucleus in humans using BrdU. Furthermore, adult neurogenesis in the human SVZ has been examined by other markers [133,151]. The progenitor cells of the rostral subependymal layer (known as the SVZ) adjacent to the caudate nucleus were co-expressed with the polysialic acid form of neural cell adhesion molecule (PSA-NCAM, a marker for migrating cells) or class III β-tubulin or Hu proteins which are specific for neuroblasts/neurons [152]. Other markers, such as the proliferating cell nuclear antigen (PCNA, a cell cycle marker), Olig2, and DCX, a marker for migrating immature neurons, were also used to determine proliferating cells in the SVZ [153]. However, Sanai et al. [154] have demonstrated that newly proliferated cells in the SVZ and RMS are present only until 18 months of age, and they are absent in the adult human brain. In addition, cells expressing DCX and PSA-NCAM are not located in the OB, but they migrate to the striatum [137]. One of the mechanisms identified to regulate adult neurogenesis in the striatum of humans is Notch-signaling, which inhibits the generation of neurons [155]. Adult neurogenesis in the striatum of rodents (mice, rat, and rabbit) and monkeys has been reported, although its function is not well known.

In marsupials, as in laboratory rodents, adult neurogenesis persists in the SVZ, and the majority of neuroblasts migrate to reach the OB [65]. In the opossum *Monodelphis domestica*, adult neurogenesis appears to have typical properties. That is, the number of newly generated neurons can be regulated by pharmacological interventions or other factors, for example, aging, which is a physiological process. It has been shown that buspirone, a partial 5-HT1A receptor agonist provides an increase in proliferation, while aging reduces the proliferation of progenitor cells [65]. Furthermore, the olfactory discrimination test performed on opossums revealed that the performance in a behavioral test is associated with the number of adult-born neurons of the OB. Opossums with a low number of newborn neurons of the OB reached significantly worse results in the olfactory-guided behavioral test [156]. This paper suggests that in the opossum, newly generated neurons are integrated into existing circuits of the OB and are required for learning and memory of new odors.

Overall, the OB is known as one of few structures of the brain that displays neuroplasticity in both eutherians and marsupials.

### 3.2. Does Adult Neurogenesis Occur in the Piriform Cortex?

Information from the OB reaches directly via the lateral olfactory tract to the olfactory cortex. The main areas of the olfactory cortex are the anterior olfactory nucleus, olfactory tubercle, entorhinal cortex, and piriform cortex. The piriform cortex is the largest region among olfactory cortical areas. Apart from the OB input, the piriform cortex also receives afferents from the amygdala and orbitofrontal cortex [157]. Nacher et al. [158] demonstrated that some cells in the piriform cortex of adult mice express DCX, indicating the presence of immature neurons. These neurons become mature, as shown by BrdU and the NeuN double labeling method [99,128]. However, further papers have highlighted that DCX-expressing cells are generated during embryonic development of the piriform cortex and express DCX for a long period (2–3 months after birth) [158,159]. During embryonic development, DCX is expressed in progenitor cells and migrating neuronal-lineage cells, while in adult brains, transient expression of DCX is present in immature neurons. Once these cells become mature neurons, which takes 2–3 weeks, DCX expression disappears.

The time of origin of DCX-expressing cells has been studied in the piriform cortex of the young adult guinea pig following prenatal BrdU injections [159]. Prenatal neurogenesis has been shown to occur from E21 to E28 in the piriform cortex of the guinea pig. A number of newly generated BrdU-labeled cells were colocalized with DCX-expressing cells in the piriform cortex of the young guinea pig, indicating that immature neurons were present over 2 months. When DCX-expressing immature neurons of the piriform cortex mature, they structurally integrate into the existing neuronal network, serving as a source of plasticity. This shows that structures involved in olfaction, from the neuroepithelium through the OB and piriform cortex, are required to reorganize neuronal networks for proper olfaction function. Adult-born neurons of the OB or the piriform cortex neurons that express immature cell markers are appropriate candidates for establishing structural and functional synapses.

Our preliminary and unpublished data demonstrated that DCX-expressing cells were distributed in the piriform cortex of young, sexually immature male opossums. DCX-expressing cells disappeared from the piriform cortex cells in sexually mature male opossums (about 7 months of age). Based on these data, we hypothesized that DCX immature neurons in the piriform cortex might be involved in sexual development, contributing to the formation of stable neuronal connections during the sexual maturation period and promoting neuroplasticity at this stage of development.

Collectively, a large body of evidence suggests that immature neurons in the piriform cortex of young animals provide a specific form of plasticity. In particular, most of these cells remain immature for more than 2–3 months and, if necessary, they reorganize their phenotype, become mature, and incorporate into the preexisting neuronal network.

### 3.3. Adult-Born Neurons of the Dentate Gyrus (DG)

Anatomically, the hippocampal formation consists of the hippocampus proper, DG, subiculum, and entorhinal cortex [47]. The DG is not included in the hippocampus proper, but the term “hippocampal adult neurogenesis” only refers to adult neurogenesis of the DG. Cells are proliferated in the hilus and subgranular layer of the DG. After proliferation, one-half of these newly proliferated cells die quickly [160]. Surviving neurons reach the granule cell layer of the DG and make the first synapses. The axons of newly generated granule neurons form synaptic connections with the CA3 pyramidal neurons. In this way, they integrate into the existing neural network and mature further [88,161,162]. It takes 8 weeks for newly generated neurons to fully mature [163].

Adult neurogenesis in the DG is conserved across mammals and declines with age in almost all investigated species (Table 1). In the DG of aged rodents, neurogenesis is still ongoing, albeit at a lower level. It has been shown that the number of adult-born neurons was considerably lower in the DG of middle-aged rats (12 months) compared to young animals [9]. In older rats (21 months), the number of adult-born neurons was reduced by 90% compared to juvenile rats. Interestingly, in some species of shrews, neurogenesis completely ceases in the DG of aged animals [74]. In the common shrew (*Sorex araneus*), the highest rate of hippocampal neurogenesis was observed 1 month after birth. The rate of neurogenesis increased in the following months of young animals and diminished at the age of 10-month-old shrews [74]. Amrein et al. found that adult hippocampal neurogenesis was absent in nine tropical bat species (*Glossophaga soricina*, *Carollia perspicillata*, *Phyllostomus discolor*, *Nycteris macrotis*, *Nycteris thebaica*, *Hipposideros cyclops*, *Neoromicia rendalli*, *Pipistrellus guineensis*, and *Scotophilus leucogaster*) throughout life [114].

In humans, newly generated cells have been first reported in the DG of old individuals (57 to 72 years) [85]. Further papers have confirmed these data, reporting that adult neurogenesis persists in the human DG from birth to old age, even in a centenarian [164]. On the other hand, several papers have demonstrated very low numbers of adult-born neurons in the DG of individuals whose ages ranged from 7 to 77 years [131]. Accordingly, the concept of adult neurogenesis in the DG of humans has been questioned, and this has sparked a debate [132,165,166,167].

A decrease in the rate of hippocampal neurogenesis has also been demonstrated in two aged marsupial species, the fat tail dunnart and Monodelphis opossum, which have been examined so far [65,66].

Despite many studies, the role of neurogenesis in the adult mammalian brain has not yet been fully elucidated. According to some research, the presence of a new pool of neurons may play an important role in hippocampal-dependent memory function, in particular, in spatial memory [168,169,170]. In laboratory rodents, spatial hippocampal memory has been widely studied using behavioral tests, particularly the water maze test (for review, see [171]). Mice with enhanced adult neurogenesis localized the hidden platform in less time than control mice in the water maze test [172,173], while mice with reduced neurogenesis had impaired learning and memory functions [174]. Further research has demonstrated that learning effects on the development of newborn neurons’ dendritic tree [175], indicating that adult neurogenesis maintains hippocampal plasticity. However, there are many papers that did not find any correlation between the number of newborn neurons in the DG and learning and memory function [176,177,178].

The paper by Tepper et al. highlighted behavior and adult neurogenesis in young opossums at the age of 6 months and aged opossums at the age of 21 months [179]. The number of newly generated cells in the DG of young opossums was almost 3.5 times more than aged opossums. Since aging is a dynamic process that takes place over time, the number of newborn neurons in the DG of opossums declines individually depending on the stage of aging. There was a 2.5-fold difference in the number of DCX-labeled cells in the DG among 21-month-old opossums. In addition, the study indicates a correlation between the number of DCX-expressing neurons and behavior. Aged opossums with high numbers of DCX cells made fewer errors, achieving high performance levels in the water maze task.

In summary, adult hippocampal neurogenesis is a common feature of mammals that exists across mammals and declines with age in almost all investigated species, including marsupials. The water maze test makes it possible to study learning and memory function in marsupials. There is only one work in marsupials indicating that adult-born neurons contribute to learning and memory. Further research is needed to understand the role of adult neurogenesis in marsupials.

## 4. Conclusions

The key difference between marsupials and eutherians in brain development is in timing. All marsupials are born with an undeveloped brain; therefore, most brain structures develop postnatally. Conversely, in eutherians born with a developed brain, intensive neurogenesis and formation of brain structures occurs during the embryonic period, with the exception of the cerebellum. Development of the cerebellum, namely the generation of granule cells, occurs postnatally; therefore, this process is a continuation of development. In both marsupials and eutherians, neurogenesis persists in restricted brain regions throughout life.

Adult neurogenesis in the DG and SVZ/OB is conserved within mammals, although very few marsupial species have been studied. The view that adult neurogenesis is limited to the DG gyrus and SVZ of brain areas has changed during the last few decades. Research on adult neurogenesis in the piriform cortex shows that immature cells could become fully functional neurons integrated into the existing network.

## Figures and Tables

**Figure 1 cells-11-02735-f001:**
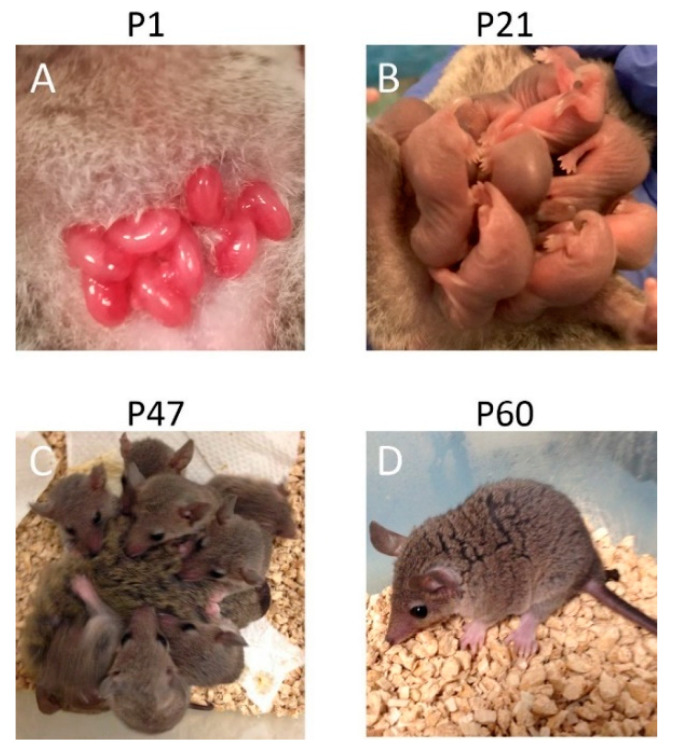
Opossums (*Monodelphis domestica*) at different ages. In (**A**), newborn opossums attached to the mother’s nipples. In (**B**,**C**), opossums at the age of 21 and 47 days remain with the mother. In (**D**), the opossum at weaning (day 60) is sexually immature. P, postnatal day.

**Figure 2 cells-11-02735-f002:**
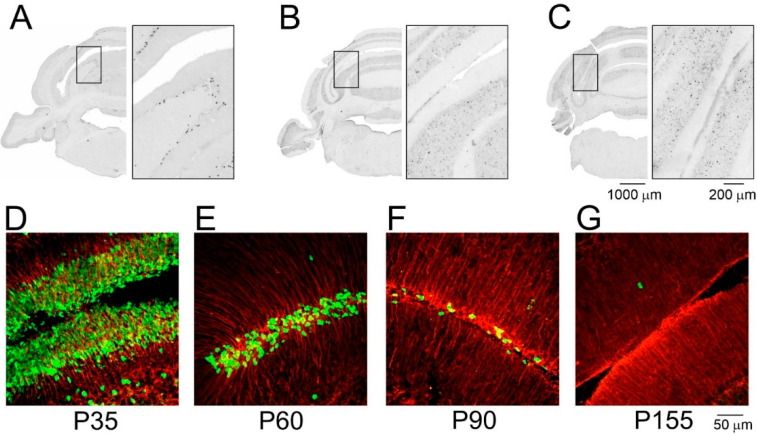
Origin of cerebellar cells in the opossums *Monodelphis domestica*. BrdU-labeled cells in the cerebellum of 3-month-old opossums that were injected at day 3 (**A**), 21 (**B**), and 60 (**C**) postnatally. BrdU-labeled cells (green) and GFAP (red) cells in the developing cerebellum that were injected with the dose of 75 mg/kg BrdU twice at postnatal days (P) 35 (**D**), P60 (**E**), P90 (**F**), and P155 (**G**). Opossums were sacrificed 4 h after the first injection of BrdU. Modified figure from our previous article [29].

**Table 1 cells-11-02735-t001:** Mammals that appear to have adult neurogenesis. AC: anterior commissure; AD: adult; AMG: amygdala; BLA: basolateral amygdala; BrST: brainstem; CER: cerebellum; CN: caudate nucleus; CP: caudate putamen; CTX: cerebral cortex; DG: dentate gyrus; EC: entorhinal cortex; EPN: endopiriform nucleus; HTH: hypothalamus; mth: month; OB: olfactory bulb; OT: olfactory tubercle; P: postnatal; PIR: piriform cortex; SN: substantia nigra; STR: striatum; SVZ: subventricular zone; TT: tenia tecta; wk: week; yrs: years.

Order/Family	Species	Gestation	Eyes Opening	Lifespan	Postnatal Developmental Neurogenesis				Adult Neurogenesis			
					OB	DG	CER	OB/SVZ	DG	CER	PIR	Other
Didelphimorphia/Didelphidae	Gray short-tailed opossum	14–15 days	P35–P37	2.5 yrs	up to P28 [2,28]		P1-P155 [29]	5 mth-2 yrs [65]	6.5–21.5 mth [65]		up to P270	
Dasyuromorphia/Dasyuridae	Fat-tailed dunnart	13–16 days	P45	1–2 yrs					4–24 mth [66]			
Didrotodontia/Phalangeridae	Brush-tailed possum	18 days	P110	13 yrs		P5–P82 [3]	up to 3 mth [4]					
Diprotodontia/Macropodidae	Tammar wallaby	25–28 days	P140	11–14 yrs	up to P25 [54]							
	Quokka wallaby	28 days	P110	8–15 yrs		P20-P85 [50]						
Euliptyphla/Erinaceidae	White-breasted hedgehog	35 days	P21	3–5 yrs				AD [80]	AD [80]		AD [80]	
Eulipotyhla/Talpidae	European mole	30 days	P22	2–3 yrs				AD [80]	AD [80]		AD [80]	
Eulipotphla/Soricidae	Hottentot golden mole							AD [79]	AD [79]		AD [79]	CTX [79]
	Giant other shrew	22 days	P20–P24	5 yrs				AD [76]	AD [76]		AD [76]	OT, EPN [76]
	Pygmy shrew, Common shrew	22–24 days	P20–P24	1 yr				1 yr [74]	5 mth [74]			
	Greater white-toothed shrew, Eurasian water shrew, African giant shrew, Asian house shrew	20–31 days	P20–P24	1.5–3 yrs				AD [75]	AD [75]			
Rodentia/Muridae	Rats (Long Evans, Sprague-Dawley, Wistar, Fischer 344, Brown Norway wild rats)	21–23 days	P13–P15	2 yrs	P0-P10 [87]	P6-P15 [6,88] up to P14 [89]	up to P21 [63]	AD [90]	6–21 mth [9,67] 8–9 wk [88]		AD [91]	HTH 2 mth [92,93] CTX, STR 9–10 wk [94] BrSt AD [95]
	Mice (C57BL/6, CD1, BALB/c, ICR, A/J, FVB, C3H/HeJ, 129/SvJ, DBA/1, DBA/2)	19–21 days	P10–P13	1–1.5 yrs	P0-P20 [53,96]	up to P20 [48,49,97]	up to P15 [55]	3–4 mth [98] 2 mth [99]	3–4 mth [98,100]		AD [101]	SN 2–20 mth [102] BLA 2–4 mth [103] CTX, STR 2–4 mth [98,104] HTH 3–4 mth [105]
Rodentia/Cricetidae	Syrian hamster	16 days	P12–P14	2–3 yrs				2.5 mth [106]	P28-P49 [107]			BLA, HTH P28-P49 [107]
	Meadow vole, Prairie vole	21 days	P14	3–16 mth				3–5 mth [108,109,110]	3–5 mth [108,109,110]			CTX, CP, BLA, THT 3–5 mth [108,109]
Rodentia/Caviidae	Guinea pig	65–68 days	Born with open eyes	4–5 yrs		up to P30 [68]		6, 12 mth [111]	1 yr [68,112]		12–14 mth [113]	
Rodentia/Bathyergidae	Highveld mole-rat, Cape mole-rat, Naked mole-rat, Damaraland mole-rat	70 days	P14	6–15 yrs					1 yr [114] 2–9 yrs [115] AD [116,117]		2–9 yrs [115]	BLA, 2–9 yrs [115]
Rodentia/Leporidae	New Zeland white rabbit	31 days	P7	9 yrs			P10 [6]			1–3 yrs [6]		CN, AD [118]
Hyracoidea/Procaviidae	Rock hyrax	200 days	Born with open eyes	8–12 yrs				AD [79]	AD [79]		AD [79]	CTX, AD [79]
Macrosce-lidea/Macroscelididae	Eastern rock sengi, Four-toed sengi	40–60 days	Born with open eyes	4–6 yrs				AD [79]	AD [79]		AD [79]	CTX, AD [79]
Artiodactyla/Bovidae	Ilede-France sheep	147 days	Born with open eyes	10–12 yrs				AD [73]	AD [73]			HTH, 18–24 mth [105]
Carnivora/Felidae	Domestic cat	64 days	P7–P10	12–18 yrs							18–24 mth [71]	CTX, EC 18–24 mth [71]
Carnivora/Canidae	Domestic dog	61 days	P10–P14	10–13 yrs					2–6 yrs [70,119]			
Carnivora/Mustelidae	Ferret	42 days	P32	5–12 yrs			P4, P21 [120]					
Scandentia/Tupaiidae	Tree shrew	46 days	P21	12 yrs				2 mth-6 yrs [78]	AD [77]		2 mth -6 yrs [78]	BLA 2 mth-6 yrs [78]
Chiroptera	Microchiroptera bats	44–180 days	P1–P2	20 yrs				AD [121]	AD [81]		AD [121]	AC, BLA [121]
	Megachiroptera bats	4–6 m	P1–P2	20 yrs				AD [122]	AD [82,122]	AD [122]	AD [122]	BrSt, Tectum, AD [122]
Primates/(New World monkey) Callitrichidae	Common marmoset	151 days	Born with open eyes	12 yrs	up to P30 [123]				1.5–7 yrs [124] 4 yrs [123,125]			CTX, CC, AMG 4 yrs [125]
Primates/(New World monkey) Cebidae	Squirrel monkey	160–170 days	Born with open eyes	21 yrs				4–6 yrs [126]	7–10 yrs [127]		3-6 yrs [128]	CTX, BLA 3-6 yrs [128]
Primates/(Old World monkey) Cercopithecidae	Rhesus monkey	165 days	Born with open eyes	30 yrs					12, 21, 31 yrs [129]			CTX, BLA 12, 21, 31 yrs [129]]
	Macaque monkey	162 days	E125	25 yrs					5–23 yrs [84]		6-12 yrs [128]	CTX AD [130], BLA6-12 yrs [128]
Primates/Hominidae	Humans	280 days	E195	75–80 yrs		3 wk-1 yr [131,132] up to 5 mth [132]		16-69 yrs [133]	23–72 yrs [85] AD [134,135,136] 14–79 yrs [135]			STR 3–79 yrs [137] AMG 24–67 yrs [138]

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
