# Peer review of "Postnatal and Adult Neurogenesis in Mammals, Including Marsupials"

_cells, 2022, doi:10.3390/cells11172735_

Round 1

Reviewer 1 Report (New Reviewer)

The work is an important contribution to the field and has been thoroughly researched.  The statements are justified.  Some of the English could be improved: several sentences are ambiguous as currently constructed.

Detailed comments:

L17. better to say "whereas, in the dentate gyrus, cells reside within this structure after division and differentiation."

L325 "subventricular zone"

Table 1 - This table might be easier to read if formatted as landscape.  There is too much crowding in the species column, with unsightly carriage returns dividing words.  There are excess parentheses in the human sections of the table.

L332 better to say "Progenitor cells leave the SVZ of the OB after division and migrate long distances."

L336 ambiguous meaning. The sentence as written implies that the thalamus is a relay structure for the olfactory system, which is not correct.

L342 generating neuroblasts which migrate.

L346 newly generated.

L350 "And, it turns out"  Although poetry and colloquial language may commence a sentence with a conjunctive, it is actually incorrect English grammar.  Conjunctives like "and" or "but" join two statements.

L360 genetically manipulated.

L424  better to say "sexually mature male opossums of about 7 months"

L426 This seems like an unwarranted conclusion.  Coincidence is not proof of causality.

L431 Large body of evidence.  Evidence is a singular noun.

Author Response

Reviewer 2 Report (New Reviewer)

This paper reviews the nature of postnatal and adult neurogenesis in mammals, including eutherians and marsupials.  It is a novel synthesis of the literature, and it is well written overall.  However, I would recommend having the final draft edited by a native English speaker, as there were quite a few grammatical mistakes.  I have indicated some of these grammatical mistakes under “minor comments”, but I likely did not catch all of these mistakes.  I also have a few more general comments, indicated below under “major comments”.   

Major comments

1. My main concern is that the paper is not consistently focused on a particular topic, and the introduction makes the function of this review fairly vague.  There seem to be two novel components of this review:  1) comparisons between marsupials and eutherians, and 2) comparisons between postnatal development and adult neurogenesis.  However, strong conclusions on these topics are not consistently provided, and I would recommend providing clearer concluding statements at the end of each section.  The paper also goes into tangential topics, not directly relevant to the comparative component of this review.  The main example of this is the fairly superficial discussion of the many factors that regulate adult neurogenesis in the dentate gyrus (lines 439-487).  Rather that considering all these factors, I would recommend expanding on the comparison of marsupials vs. eutherians (lines 488-510).

2. On page 2 (lines 61-65), it is suggested that whether or not adult neurogenesis occurs in humans is “related” to whether or not neurogenesis occurs in other brain regions.  It is not obvious how these two topics are connected, so this should be explained or reworded.

3. The tammar wallaby and the short-tailed opossum are used as two key examples on page 3.  It would be useful to indicate the relative adult body size of these two species for comparisons with rats and mice.  A few sentences about the ecology of these species would also seem useful.  What is their natural diet and habitat?

4. Lines 183-198:  This paragraph provides a comparison of the quokka to the rat.  However, it is not clear what conclusions can be drawn from this comparison.  Similar to my comment #1, it would be useful to have some speculation regarding the functional significance of inter-species differences in developmental timing.  As a more minor point, it is not immediately obvious that the “short-tailed scrub wallaby” and the “quokka” are the same species.  I’d suggest consistently using one of these two common names.

5. Lines 222-234:  This concluding paragraph is vague and wordy.  It would be much more interesting to provide some direct comparisons of the developmental timing of neurogenesis in eutherians and marsupials.  Some speculation regarding the functional significance of the different developmental trajectories would also be interesting.

6. Table 1 is quite useful, but the formatting is poor.  The columns are too close to each other, making it difficult to distinguish regions involved in postnatal or adult neurogenesis.  I’d suggest shifting the table so that it is in landscape format across the page.  Additionally, there is a heading indicating that “Order/suborder” are reported, but in fact Family names are often listed.  I would suggest listing both Order and Family or all groups for consistency.

7. Line 410:  It is stated that piriform cortex expresses DCX for a “long period”.  It would be useful to be more specific here to allow direct comparisons between the embryonic period and adult neurogenesis.

8. Line 465:  It is stated that “The secretion of steroid hormones is mostly associated with a stressful situation”.  This is only true for glucocorticoids, rather than all steroids.  As mentioned previously, this entire paragraph, discussing the effects of steroids on neurogenesis could be deleted, as it is not one of the central topics reviewed in this paper.

9. Line 513:  It is stated that “Based on studies it becomes obvious that in marsupials, adult-born neurons contribute to learning and memory”.  This seems to be an over-statement, as this conclusion is based upon only one published experiment.  The degree to which adult neurogenesis is needed for learning and memory in rodents remains quite controversial, and many experiments have been performed with rats and mice.  So, suggesting that one experiment is definitive for marsupials seems inappropriate.  Additionally, it would be useful to provide a more direct comparison in this section on the effects of aging on rodents vs. marsupials. 

Minor comments

1. Line 32:  “Now, it is well established” should be “It is now well established”.

2. Line 43, remove the comma:  “marsupials, and the postnatal” should be “marsupials and the postnatal”.

3. Line 49:  “Adult neurogenesis is demonstrated” should be “Adult neurogenesis has been demonstrated.”

4. Lines 53-54:  “New approached including 5-bromo-2-deoxyruine” should be “New approached including using 5-bromo-2-deoxyruine”

5. Lines 75-76:  delete the word “Nowadays”.

6. Lines 88-89:  “through which newborn opossums have to travel and attach” should be “used by newborn opossums to travel and attach”.

7. Figure 1 caption (line 113):  “In (D), the opossum at weaning (day 60).”  This is an incomplete sentence, needing a verb.

8. The species name “Monodelphis domestica” is used repeatedly.  It is customary to abbreviate the genus after the first time a Latin species name is mentioned.  In this case, “M. domestica” should be used after the first time the species name is used.

9. Line 129:  “The mice gestation” should be “The mouse gestation”.  This same mistake is made multiple times throughout the paper.  “Mouse” should be used as an adjective rather than “mice”.

10. Lines 136-137:  delete “which are located above the optic chiasm”.  This is obvious based on the name “suprachiasmatic”.

11. Lines 145-146:  “place within 6 days” should be “place over 6 days”.

12. Line 156:  “eutherians developed a new commissure” would seem to be more accurately described as “eutherians evolved a new commissure”.

13. Line 159-160:  “organization; it is extremely” should be  “organization; it is extremely”.

14. Line 187:  “Angevine [48] has demonstrated” should be “Angevine [48] demonstrated”.

15. Line 190:  “neurogenesis in developing mouse brain” should be  “neurogenesis in the developing mouse brain”.

16. Line 192:  “However, a research” should be “However, research”.

17. Line 210:  “Although mitral and tufted are generated” should be “Although mitral and tufted cells are generated”.

18. Line 214:  “becomes apparent visible after” should be “becomes apparent after”.

19. Line 316:  “which cells express” should be “where cells express”.

20. Line 334:  “and locate in the olfactory bulb” should be “and mature in the olfactory bulb”.

21. Line 350:  “mitral cells activity” should be “mitral cell activity”.

22. Line 352:  “improving the power” should be “improving their power”.

23. Line 367:  “Ericksson et al. [85] have described” should be “Ericksson et al. [85] described”.

24. Line 397:  “eutherian and marsupials” should be “eutherians and marsupials”.

25. Line 408:  “and this was shown by BrdU” should be “as shown by BrdU”.

26. Line 418:  “for the proper olfaction function” should be “for proper olfactory function”.

27. Line 431:  “a large number of evidences” is not grammatically correct.  This sentence needs rewording.

28. Line 460:  “estrogens are caused an increase” should be “estrogens cause an increase”.

29. Line 502:  “The number of new generated cells” should be “The number of newly generated cells.”

30. Line 523:  “that occur adult neurogenesis” should be “that have adult neurogenesis”.

Round 2

Reviewer 2 Report (New Reviewer)

This is a much improved review article, with the authors adequately addressing most of my concerns.  I would recommend some final editing for grammar prior to final submission of the manuscript.  Here are the remaining mistakes that I noticed:

1) Line 70:  "specifically focus" should be "specifically focusing".

2) Line 212:  The sentence starting "Although the body weight" is an incomplete sentence.  Additionally, the phrase "twice smaller" should be "half the size".

3) The new paragraphs on lines 213-222  and lines 252-268 need some citations to support this new information.

4) Line 262:  delete the comma:  "eutherians, is linked" should be "eutherians is linked"

5) The formatting of the table is better but should still be improved.  In particular, the column headings are squished together, with words broken in strange ways.  For example, "Lifespan" has a dangling "n" on a second line.

6) Line 451:  "DXC" should be "DCX".

7) Line 525:  "In laboratory rodents, the spatial hippocampal memory" should be "In laboratory rodents, spatial memory".

Author Response

This manuscript is a resubmission of an earlier submission. The following is a list of the peer review reports and author responses from that submission.

Round 1

Reviewer 1 Report

The manuscript by Bartowska et al., entitled „Postnatal neurogenesis in mammals, including marsupials: continuation of developmental and adult neurogenesis” is a review article. The key aims of this review are to evaluate how many proliferative neurogenic zones are present in the brains of adult mammals, including marsupials, to debate the significance of newly born neuron numbers in the dentate gyrus and the hypothalamus, and to determine whether their low number is essential for the functioning of a given area. Adult neurogenesis is an exciting topic as it represents a remarkable and unique form of plasticity. Reading the abstract of the paper, I was really curious to learn more about adult neurogenesis in previously understudied taxa like marsupials, how high its rate is compared to the usual suspects (rats or mice), and how the new neurons affect the behavior of these animals.

The review is structured into seven main sections, covering the following topics: postnatal neurogenesis in marsupials, postnatal neurogenesis in the cerebellum, proliferative zones of the brain and methodological issues, quantification of newborn neurons in the dentate gyrus and the hypothalamic region around the third ventricle. The first part of the manuscript introduces developmental neurogenesis in marsupials and adult neurogenesis in eutherians, focusing on the dentate gyrus (DG), subventricular zone/olfactory bulb and the hypothalamus, and on the cerebellum as a unique structure of eutherians that develops postnatally. I´m wondering why the authors neglect the DG, which also shows a protracted development (2-3 weeks in mice and rats; e.g. Bayer 1980; Altman & Bayer 1990; Li & Pleasure 2013) with up to 80% of the granule neurons being formed after birth. While the introduction provides an overview on all that, it does not identify the knowledge gaps or open questions from which the authors derive their aims. I would appreciate if the authors could illustrate how their review will advance the field and explain why they focus on the DG and the hypothalamus.

Unfortunately, the content of the manuscript not meet the aims mentioned in the abstract and the introduction. Regarding the first aim (“evaluate how many proliferative neurogenic zones are present in the brains of adult mammals”), I expected that beside the classical neurogenic niches (SVZ and DG), authors will include at least a short discussion of other neurogenic regions under debate, including the amygdala, striatum or hypothalamus. Even if this is a highly debatable topic and it´s unclear if the new neurons found there arise from local precursors or from outside, they should be mentioned and discussed. The focus on the hypothalamus as one of those atypical niches is commendable but appears arbitrarily. The authors include the opossum and Australian tammar wallaby as representatives of marsupials, and rats and humans for eutherians, without justifying this selection (evolutionary, based on availability of data, …?) and in an inconsistent way in the different sections. Regarding the aims related to the functional significance of adult neurogenesis, one could expect a detailed discussion of experimental and computational studies dealing with this topic. However, this is almost completely missing despite a short paragraph on newborn hypothalamic neurons and despite the extensive data available for the DG. It would be interesting to compare the neurogenesis rates in different species and link these rates to specific functions or even species-specific behaviors. Even if data for some of their model organisms might be sparse, the authors could speculate about such correlations. To facilitate this discussion, I think the authors should reorganize the content of the entire manuscript (for example: instead of organizing sections according to species, they could organize them according to brain regions and then compare the neurogenesis rates between species as well as their correlation to specific brain functions). I also suggest to change the title of the manuscript. From the current version, I expected a critical discussion of the controversy on whether adult neurogenesis is a continuation of development or an independent process, which is an intensely debated topic in the field right now.  

Following the introduction, authors continue with a section about postnatal neurogenesis in marsupials. After an introduction into the systematics of theria and the differences between eutherian and marsupials, they describe the features of newborn marsupials and their brain development, setting the focus on the opossum and tammar wallaby. While this is interesting, the purpose of this section is unclear within the defined scope of the review. It would be far more interesting and logic, if postnatal neurogenesis in marsupials would be compared with prenatal neurogenesis in eutherians, e.g. to elaborate if and how the highly immature birth and the extrauterine development affects the timing of neurogenesis and brain morphogenesis. Moreover, while focusing on the hypothalamus, the neocortex and OB, the DG is again neglected. This is unfortunate for several reasons: The DG is one of the two classical sites continuing to produce neurons throughout adulthood, and at least in rodents most of it develops postnatally, and the focus of later parts in this review is set on the DG.

Minor points in this section:

  1. The reference [21] in line 93 contains none of the information mentioned in the sentence before, despite the age of rats equivalent to the birth date of the opossum. Moreover, this study reports that the birth date of opossums corresponds to E15 instead of E14 in rats, and citing an original paper (Valverde et al., 1989). I suggest to cite this paper instead and to include references for the other statements.
  2. I suggest to provide consistent information for opossums and wallabies (e.g. gestation times) which will facilitate a direct comparison between them.
  3. The sentence in lines 139 f. should be rephrased. It becomes not clear, if authors summarize the content of the previous section (marsupials) or if they prepare for the next sections and which 3 structures they mean.

I have also concerns regarding the next section, which deals with postnatal neurogenesis in the cerebellum. Providing the cerebellum an extra section was based on the allegation that the cerebellum of eutherians is unique regarding its protracted development into postnatal periods, neglecting the similar time course in DG development, at least in rodents.

Minor points are:

  1. Phrasing in line 178: “Postmitotic cell divisions continue at a high rate up to the first 5 postnatal months and gradually the number of newly proliferated cells declines in the following months”
  2. Line 180 ff.: It does not become clear, why the authors switch to lagomophs. Why is it special? What´s the relation to rodentia to whom they are compared here? Because they are relatively closely related to rodentia but show surprising and unexpected structural plasticity (protracted neurogenesis)?
  3. What´s the difference between sentences in lines 183 ff.: “In the cerebellum of this species, newly generated neurons were detectable up to the 5th postnatal month. Interestingly, newly generated neurons were observed in the cerebellum of 3-year-old rabbits.”? Which species? Citations are lacking.
  4. Source of Figure 2 is unclear. I guess these are original images from the authors, as they are related to figures in [23]. Please note.
  5. Line 200 f.: Why suddenly mice?

The next section is named “Proliferative zones of the brain and methodological issues”. I suggest to change the header of this section to meet its content. This is a more general section providing an overview on adult neurogenic zones, the species/taxa in which they have been described and a short overview on the markers applied to study adult neurogenesis. First, the DG and SVZ neurogenic niches are very shortly described, followed by a paragraph and table summarizing species displaying adult neurogenesis, mostly focussing on eutheria. This table appears somehow inconsistent and could be clearer. It lacks examples of marsupialia despite the opossum. Are there no data for the tammar wallaby? If so, authors could include data from other pouched marsupials (e.g. Harman et al., 2003) and mention in the text that data on adult neurogenesis in marsupialia is sparse. The table also lacks the New Zealand white rabbit, which has been mentioned in section 3; instead, it shows data of the European rabbit. It would be nice if the authors could appraise these data, e.g. to identify common traits or differences between taxa. Without such a synthesis, this table appears somehow expendable.

In the part discussing the markers applied to quantify neurogenesis, a more detailed discussion of their suitability, advantages and disadvantages would have been desirable. This is highly important, as the use of inappropriate markers may induce substantial bias in an analysis. Thus, BrdU is best used for studying the fate of newly born cells (differentiation or survival, the latter only if including at least two chase intervals with one immediately after the BrdU pulse) or for studying cell cycle dynamics in combination with an endogenous proliferation marker. To study proliferation rates, endogenous proliferation markers are best suited. Cell-specific markers are also applied to quantify newly born cells/neurons, however, resulting data have to be treated cautiously, as none of the neurogenic lineage markers is specific for just one cell type.

Minor points:

  1. Please add a reference to your statement in line 224 f.: “The rate of adult neurogenesis in the dentate gyrus is lower than in the SVZ.” This is stated several times without any reference.
  2. Lines 258 ff.: The meaning of these sentences is unclear. Please revise and add a reference.

I have also concerns regarding the section “Quantification of newborn neurons in the dentate gyrus”, which is divided in subsections for rats, humans and the opossum. It would be far more straightforward and interesting, if the authors would directly compare the species/taxa selected, marker by marker, instead of organizing the sections according to species. In particular, if we keep the aims of the review in mind. There is no integration of the data reviewed in this section. Authors could debate the neurogenesis results from different species, speculate why there are similarities or differences between species or why the human AHN is so difficult to study. Furthermore, in my opinion it would be more purposeful to discuss studies using similar markers to compare neurogenesis rates between species or brain regions.

The subsection on rats is neither goal-directed nor straight. It lacks seminal papers and mixes up data from different BrdU regimes. In general, a comparison of papers examining endogenous proxies of proliferation and new neurons would have been more appropriate to compare the rates of neurogenesis between studies, ages and species. It´s not clear why this section focuses on BrdU, which, as the authors correctly explain in the section before, has technical limitations regarding the comparability of data from different studies. For rats (and mice), studies exist for almost every endogenous marker and age, therefore, it is incomprehensible to me why the rat section focuses on BrdU. I agree that citing BrdU studies is important to compare differences in the potential/differentiation of new cells, but this is not covered in this section. Discussing studies using endogenous markers would also facilitate the comparison with human data in the next subsection, as we have just the seminal study from Peter Eriksson using BrdU, while there is plenty of data using Ki67 and other endogenous markers (DCX, PSA-NCAM). The choice of rats over mice as representative for rodents is also questionable, as there exist by far more data in mice. Finally, this subsection appears as mere collection of information, missing a critical debate and a discussion of the functional relevance.

Other points are:

  1. Line 277 f.: Authors state “The literature search did not yield another work with such BrdU-labeled cells in the dentate gyrus of adult Sprague-Dawley rats”. I was surprised by this statement because several such reports exist, including the studies of Heather Cameron´s group. In their comprehensive study published in 2009 (Snyder et al., J Neurosci) they analysed similar ages, applied several protocols of which some also match the studies mentioned in this review, and they compare mice vs. Spargue Dawley rats.
  2. It´s not clear why the authors start to compare males and females.
  3. Line 289: Instead of reporting the weight authors should mention the age of rats to better compare with other studies cited.
  4. Line 290 ff.: Please revise the text and the reference. The cited paper did not apply the injection protocol mentioned here. They applied a bolus of 300 mg/kg BrdU.

The next subsection describes adult neurogenesis in humans. Here, it is noticeable that the authors describe many technical details of the studies cited. To be more consistent with the other sections and to improve readability, I suggest to focus on the data. Instead, a debate on the technical reasons for the controversial human data would be welcome at the end of this section. Moreover, up to this point the authors focused on neurogenesis in young adults. I´m wondering why they now focus on aging and data from old people or even don´t differentiate by age in studies comprising multiple age groups.

Other points are:

  1. Line 341: Here they could also include the study from María Llorens-Martín, which appears in the reference list but not linked anywhere in the text. Same for Spalding paper.

In the subsection summarizing data in opossums I suggest to revise the conclusion at the end in a way that it meets the scope of the manuscript (lines 375 ff.).  Here, authors also undertake a discussion of aging. If this is their aim, I would do this consistently, also in the rat section where it´s lacking. Moreover, the cited papers evaluated the spatial learning of opossums in the Morris water maze. I´m wondering why the authors didn´t comment on that or discussed that in a section about the functional relevance of adult neurogenesis.

There are several points that need to be revised in the section focusing on the hypothalamus, which presents data obtained in rats and mice. In subsection “The third neurogenic zone located in the hypothalamus around the third ventricle” I suggest the following:

  1. Lines 387-392: Please revise. Argumentation is unclear and misleading.
  2. Conclusion in lines 400 f. seems to ignore citations [144] and [145]. Please comment on why you think that [145] concluded that, in vivo, the mitotic power of the hypotalamic progenitors is inhibited?
  3. Lines 417 ff.: “However, a previous review paper by Yoo and Blackshaw [151] reported that tanycytes are only candidate stem/progenitor cells and concluded that further experiments are needed to clarify this view.” That´s an interpretation of these authors in a review, but they didn´t contribute original data. It would be desirable if authors draw their own conclusions instead of repeating statements from other reviews.
  4. Please revise phrasing in lines 429 f.: “Hypothalamic neurogenesis regulating the energy balance, influences body weight.”

In subsection “Quantification of proliferating cells in the hypothalamus”, different ages are mixed throughout in a non-systematic manner, also including data covering the postnatal period. More specific points are:

  1. Hu as a marker should be introduced to facilitate the interpretation of the data presented.
  2. Line 475: I suggest to add a discussion of the data presented before.
  3. Lines 476 ff.: Paragraph starts with aging, but no corresponding data follow (no comparison of data from 4 mo. and 23 mo. old mice) and no conclusion. Instead, authors give an imprecise statement on the outcome resulting from different BrdU application paradigms.
  4. Lines 478 ff.: “After 4 weeks of infusion mice were perfused and the distribution of BrdU-labeled cells was defined 8 weeks later, after treatments.” How does that work?
  5. Line 491 f.: “might be crucial for controlling body homeostasis and the regulation of physiological functions.”. I expected a discussion of functional role following.
  6. The text following thereafter is confusion and the argumentation is incomprehensible.

For the conclusion of the entire manuscript, I suggest to synthesize the information presented, identify gaps and derive new ideas or perspectives for future research.

Finally, there are some general points that should be addressed:

  1. Language and spelling needs to be revised, e.g.

-              Please remove unnecessary spaces throughout the manuscript (e.g. lines 20, 207, 366, 386)

-              Please revise phrasing, e.g. lines 138-139: it becomes not clear if the authors summarize the content of previous section (marsupials) or if they prepare for the next sections; which 3 structures?

-              Line 33: “[for review, see [1].” There is a ] missing.

-              Line 55: Please exchange “compar” by “compare”.

-              Line 208: remove “of the rabbit”

-              Line 84: Opossum is not equal to possum

-              Please check if all abbreviations are introduced at first appearance, e.g. RMS in line 216

-              Line 268: Please delete “ubsection”.

-              Line 286: Please exchange the “,” by a “.”.

  1. Please revise citations.

-              Several papers occurring in the reference list are not cited in the text.

-              Authors could cite more recent publications.

Reviewer 2 Report

In this review by Bartkowska et al., authors elegantly summarized what is known about neurogenesis in mammals that occurs postnatally and the several proliferative neurogenic zones present in the brain of adult mammals including a debate on the significance of the function of newborn neurons in the dentate gyrus, the SVZ/OB, and hypothalamus.  For a long time, it was thought that during adult life continual production of new neurons, a process is known as adult neurogenesis, occurs only in two mammalian brain areas: dentate gyrus and SVZ, in this review the authors were able to also describe the new discoveries about adult neurogenesis in a new brain region: the hypothalamus.

The review is well-written and easy to comprehend. The figures and tables included are nicely delineated and explained in context.

This review does a nice job of combining all that we have learned about postnatal neurogenesis including adult neurogenesis and how regulates its function. I think this is a nice addition to the neurogenesis literature.

Author Response

We thank the reviewer for supporting our work.

Round 2

Reviewer 1 Report

I appreciate that the authors considered the majority of my previous comments and tried to address them to improve the manuscript. Still, there are several important points that need to be improved.   

1.      Although goals are made more clear now, authors still state that, as mentioned for example in the abstract, they will debate “the significance of the number of newly born neurons in the dentate gyrus and the hypothalamus., and whether their low number is essential for the functioning of a given area”. However, there is no such debate for the dentate gyrus, neither in rodents nor in an integrative manner considering all the taxa discussed in this review.

2.      Also, a comparison of data from dentate gyrus and hypothalamus, as announced in the introduction (“We look at the number of new neurons localized in the dentate gyrus and the hypothalamus, compare these data, and show how and whether they are relevant to the functioning of a given brain area”) is lacking. I understand that this is difficult given that mostly BrdU data are presented which result from different labeling regimes, chase periods and quantification strategies. To circumvent that, either the aims should be appropriately defined or studies applying better comparable methods should be discussed.

3.      I suggest to revise how the sections were “reorganized”. Basically, the authors just changed the headers which introduces more inconsistencies and confusion. Now, we have a section 3 “Adult neurogenesis in mammals” followed by section 4 “Quantification of newborn neurons” comprising a discussion of the techniques used to study adult neurogenesis but also the data for the DG. This is followed by a section 5 “The hypothalamic region around the third ventricle”. What about a section 3 “Adult neurogenesis in mammals” with 3.1. “Methods to quantify adult neurogenesis in mammals”, followed by 3.2 “Quantification of neurogenesis in the DG” with subsections for opossum (3.2.1), rat (3.2.2) and human (3.2.3), and 3.3 The hypothalamic region around the third ventricle” with corresponding subsections.

Still, the discussion of the functional significance and a comparison between species is lacking.

4.      I have still major concerns regarding the focus on BrdU in sections aimed to evaluate neurogenesis rates of different taxa/brain regions. While I fully understand that the authors want to focus on similar markers to facilitate comparability, BrdU is the least useful one for as long as injection regimes, chase periods and the analysis strategy aren´t the same in the studies discussed. This is not the case. Taken that BrdU is stated to be the only marker applied in studies on Marsupials, focusing on it for all other species comes at the cost of comparability between all of them.

The manuscript would largely benefit if authors would focus only on such studies that applied highly similar BrdU regimes and readouts for comparing neurogenesis rates in the same brain region of different species, at best such that report on total numbers in the entire structure or on densities.

This does not preclude that studies using different protocols may be cited to discuss similarities and differences in neuronal differentiation of newborn neurons (e.g. %NeuN+BrdU+ in all BrdU+), as this just depends on the post-injection survival period.

If available, studies reporting on total numbers or densities of proliferating cells, neuroblasts and newborn neurons should be implemented – again focusing on such markers that have been applied in the majority of species and brain regions to be compared (e.g. Ki67, DCX, PSA-NCAM: data are available for neurogenic brain regions in rats, mice, humans) or on such providing comparable information (e.g. Ki67 and MCM2 as proxies of proliferating cells).

5.      The section “Quantification of newborn neurons in the dentate gyrus…” (rat part) has not been extensively revised. There is neither a correction of statements that are obviously wrong nor was it tried to get rid of the inconsistencies throughout.

a.       lines 369ff.: “The literature search did not yield another work, in which the same dose of BrdU was injected to 2-month-old with such BrdU-labeled cells in the dentate gyrus of adult Sprague-Dawley rats” This statement should be revised as there is at least another study on same rat strain and with same study design in one of their groups (Snyder et al., 2009).

b.      Related to the point 4, there is a mix of BrdU doses and injection regimes, survival times, analysis strategies (total numbers, numbers per section, numbers per mm2) and ages. Also, it needs to be considered that numbers per section and per area depend on section thickness, which makes these data even less comparable.

6.      As suggested, authors removed unnecessary technical details from the section focusing on humans (lines  393ff.). A critical synthesis and debate of the seemingly conflicting data is still missing, as is a discussion of functional implications.

Moreover, I have concerns about the conclusion in lines 442ff.. Even if new methods to detect neurogenesis in humans will help to provide more valuable information, lack of techniques per se is not the main issue (e.g. we have data from BrdU-injections, endogeneous markers detected by immunolabeling, MR-spectroscopy, radiocarbon detection). The problem lies in the accessibility of tissue material of standardized quality and the standardization of techniques, as has been convincingly shown by the group of Maria Llorens-Martin. This should be discussed.

7.      The hypothalamus section (lines 542ff.) needs still a revision to improve clarity and readability.

a.       Lines 580ff.: As stated in my previous report, the paragraph starts with aging, mentioning a paper studying 4 mo. and 23-24 mo. old mice [181], but the corresponding data are not there. Please clarify to which age data in lines 585ff. relate and add data for both ages. Otherwise this paragraph is redundant.

b.      Please arrange the sentences in a coherent way, I guess the sentence in line 588 “The number of BrdU+ cells decreased in aged mice” belongs to reference [181]?

c.       What is the purpose of this sentence: “Less BrdU-labeled cells were observed when 25 mg/kg BrdU was ip injected every 2 h for 2 days.” Do authors want to show that different BrdU paradigms result in different number of labeled cells? Is that necessary if cell numbers are not mentioned? Please cite the source of this data.

d.      Please also revise the new sentence on AraC treatment in lines 588ff. It appears misplaced and may be included in the functional discussion starting in line 604. Reference is lacking.

e.       The paragraph starting in line 591 has not been revised and the argumentation is still incomprehensible. It is also unclear to the referee, if authors want to claim in the last lines 600ff. that BrdU+ cells are neurons, regardless of almost none of them express the neuronal markers Hu and NeuN?

8.      The manuscript still requires careful language editing.  

Minor points:

9.      In the new paragraph on the development of the dentate gyrus in rodents, please consider that for the dentate gyrus the majority of DGCs are born during the first 2 postnatal weeks (in mice & rats; first week is the peak, e.g. Avenigne 1965 and Bond et al., 2020) and does not cease at P4. Interesting to note in perspective of a more integrative view, neuron production for the nascent DG starts around the time when neocortical neurogenesis begins (Avenigne 1965 and Bond et al., 2020; Hevner et al., 2003) but lasts much longer than the latter.

Furthermore, this passage can be condensed. Some sentences contain no information (e.g. line 179ff.) or repetitions. Several times, studies are mentioned without reporting the information they provide within the scope of the review (e.g. lines 179ff. or 181f.).

10.   Please check references regarding stage correspondences again and revise the data in the text (Valverde et al., 1989; Cardoso-Moreira et al., 2019)

11.   Please provide consistent information for opossums and wallabies in section 2, still the gestation time for the opossum is missing, instead the weight at birth is given (lines 118f.).

12.   Line 256: “first and the only species investigated so far that shows adult cerebral neurogenesis”. Please mention the category of which the rabbit is the only species and exchange “cerebral” by “cerebellar”.

13.   The new passage “We compared the brain development of opossums to that of mice… However, adult mice weigh 3.5-4 times less than opossums.”(now in lines 133-138), appears misplaced and interrupts the text on marsupials. Please shift it more upwards, i.e. to the position you mentioned in your response letter.

14.   Sentences in lines 260ff. should be re-phrased to prevent repetition, for example to “In the cerebellum of this species, newly generated neurons were observed until an age of at least 3 years [6]. At this…”

15.   Lines 349ff.: Sentence “Since the rate of adult neurogenesis is lower in the dentate gyrus than in the SVZ, we reviewed the papers that reported the numbers of labeled cells, in order to further assess whether a low number of newly produced neurons is relevant to a given brain structure.” should have been revised according to the authors response letter, but it wasn´t. Please specify in which context the first statement applies (with a reference).The reasoning remains unclear.

16.   New sentences in lines 471ff.: Please specify what kind of cells you refer to (BrdU+?). The mix of “cells” and “nuclei” is a bit confusing. If this information relates to BrdU labeling please use one or the other. Please provide the temporal context – are these numbers produced per day?

17.   Sentences like “The phenotypes of BrdU-labeled cells were also determined using double immunohistochemistry” are redundant and should be removed, if the corresponding data aren´t mentioned.

18.   Line 411: Please explain what are “control patients”.

19.   Please revise text in lines 486-494.

20.   Lines 487f.: Please exchange “immature mitotic neurons” by the correct term, e.g. neural progebitor cells. Neurons do not divide.